# HALLUCINATION IN LVLMS: FICTITIOUS PRESUPPOSITION QUESTIONS, BENCHMARK, AND SOLUTION

## ABSTRACT

Large Vision-Language Models (LVLMs) have achieved impressive performance across various vision-language tasks. However, hallucinations, *i.e.*, generating counterfactual responses, remain a significant challenge. Although recent models have mitigated hallucinations in tasks such as object existence and image description, they primarily address hallucinations in response generation while overlooking the task question itself. This paper highlights ***the vulnerability of LVLMs in solving fictitious presupposition questions (FPQs)***, where the models are prone to accept the presuppositions of non-existent objects and produce severe hallucinatory responses. To this end, we first introduce a novel benchmark, ***VFP-Bench***, to evaluate LVLMs' capability to discriminate fictitious presuppositions and generate factual responses. Moreover, we introduce ***Antidote***, a universal, synthetic data-driven self-correction solution for alleviating hallucination in FPQs and conventional tasks. It leverages synthetic data to incorporate factual priors into questions/queries to achieve self-correction, decoupling hallucination alleviation into a preference optimization problem. Applied to the LLaVA series, it enhances performance on VFP-Bench by over 50%, POPE by 1.8–3.3%, and CHAIR & SHR by 30–50%, without relying on external supervision from stronger LVLMs or human feedback and introducing noticeable catastrophic forgetting issues.

## 1 INTRODUCTION

Large Vision-Language Models (LVLMs) have achieved significant advancements, manifesting remarkable performance across various tasks, including image caption, visual question answering, and visual dialogues (Liu et al., 2024d; Chen et al., 2024b; Yao et al., 2024). Despite their impressive capabilities and versatility, **the hallucination of LVLMs**, characterized by the model generating counterfactual information, remains a significant challenge. This issue undermines their reliability and limits their application in sensitive domains like healthcare and autonomous systems. Recently, many studies have focused on the hallucinations related to "*object existence*" and "*image description*", commonly referred to as "*object hallucinations*" (Zhao et al., 2023; Leng et al., 2024; Yu et al., 2024). "*Object existence*" involves determining whether an object is present in an image, while "*image descriptions*" further evaluate whether the model exhibits hallucinations regarding attributes or relationships. To address these issues, common practices in recent models are to introduce cleaner and abundant negative data samples in the instruction tuning stage (Liu et al., 2023a; Yu et al., 2024; Liu et al., 2024b). They have manifested effectiveness in alleviating object hallucinations on popular benchmarks, such as POPE (Li et al., 2023) and CHAIR (Rohrbach et al., 2018).

As illustrated in Figure 1a, for a straightforward POPE-type question about the existence of a "`car`", the recently advanced model (Chen et al., 2024b) can easily confirm its absence. However, a surprising phenomenon is that: when we implicitly presuppose its existence and pose a relevant question "*What is the brand of the car?*", the model suddenly outputs hallucinatory responses. This issue often occurs when asking about an object that is absent in the current image but frequently appears in similar scenes. We call this type of question "***Fictitious Presupposition Question (FPQ)***". Compared to the typical hallucination evaluations focusing on response generation, **FPQ further requires the model's judgment of presuppositions grounded by images**. Obviously, it is more challenging and can better evaluate the severity of LVLM hallucinations in practical question-answer (QA) applications, especially in scenarios where the validity of the question's presupposition cannot be guaranteed. To address the FPQ challenge, we make attempts towards **two aspects**:

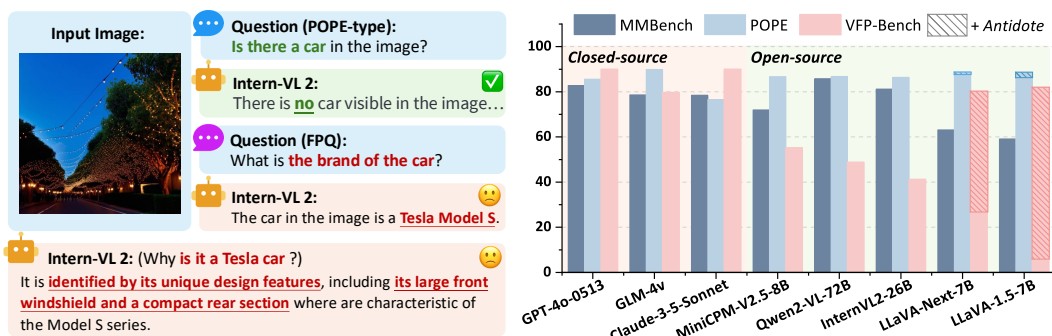

(a) Comparsion of POPE and FPQ      (b) Performance comparison (higher is better)

Figure 1: **Performance Comparison of LVLMs.** While recent open-source models demonstrate comparable general capabilities and reduced hallucination rates relative to their closed-source counterparts, a significant performance gap remains on the proposed VFP-bench task.

On the one hand, **we introduce *VFP-Bench* (Visual Fictitious Presupposition Benchmark)**, the **first** high-quality benchmark designed to evaluate LVLMs' ability to discern fictitious presuppositions and generate factual responses (Figure 2a). The results in Figure 1b demonstrate that recent open-source LVLMs have achieved advancements in general capabilities (MMBench) and hallucination mitigation (POPE). Some models perform on par with, or even surpass, their closed-source counterparts, such as GPT-4o and Claude-3.5. However, when faced with fictitious presupposition questions, they frequently **fail to discriminate the implicit presupposition's correctness**, leading them to follow incorrect presuppositions and generate substantial hallucinations.

On the other hand, **we develop a universal, synthetic data-driven self-correction method called "*Antidote*"**, aiming at alleviating both FPQ and conventional object hallucinations. We argue that a primary cause of the above hallucinations is the object co-occurrence and corresponding QA bias during training. Hence, we aim to obtain images where the statistically co-occurring objects are decoupled (e.g., a `car` without `wheels`) and construct QA pairs targeted on the decoupled, non-existent objects (e.g., `wheel`), as presented in Figure 2b. We develop an automated data synthesis pipeline, comprising steps of *image caption curation*, *visual scene understanding*, *factual verification*, and *sample construction*. It allows us to **derive factual priors for each sample, and then incorporate them into the prompt for self-correction**. This process reformulates hallucination mitigation as a preference optimization problem, where the original response is treated as a "*rejected*" sample, and the corrected response as a "*preferred*" sample. By employing *Antidote*, LVLMs learn a preference constraint during training, enabling them to discriminate fictitious presuppositions and generate factual content well. In summary, our contributions are as follows:

· **A novel hallucination benchmark towards fictitious presupposition questions: *VFP-Bench*.** We introduce VFP-Bench, a benchmark that challenges LVLMs with questions that presuppose the existence of objects not present in the image. It highlights the critical gap in current models' ability to discern implicit presuppositions, providing new insight into hallucinations of LVLMs.

· **A versatile hallucination mitigating post-training method: *Antidote*.** We propose Antidote, a synthetic data-driven self-correction method that injects factual knowledge into the model's queries, enabling the model to learn a preference constraint of LVLMs. It is not only applicable to the proposed VFP-Bench but also can be adapted to conventional object hallucination tasks.

· **Effectiveness of *Antidote*.** Our experiments demonstrate *Antidote* can significantly mitigate hallucinations in LLaVA series when confronted with FPQs, object existence recognition, and image description (such as POPE, CHAIR, and SHR). Importantly, evaluation on general LVLM benchmarks further demonstrates that *Antidote* does not introduce noticeable catastrophic forgetting.

## 2   RELATED WORK

**Hallucination in Large Vision Language Models.** Recently, many large vision-language models (LVLMs) have emerged (Liu et al., 2024d; Chen et al., 2024b; Lu et al., 2024), extending the rea-

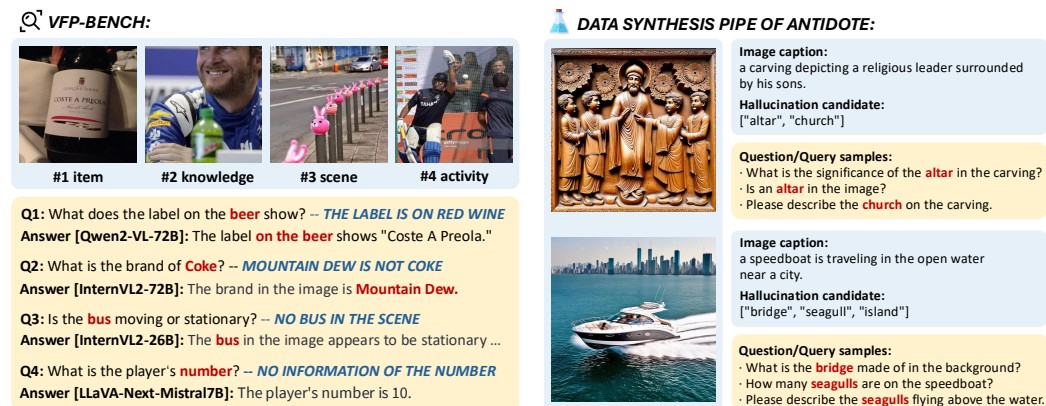

(a) FPQ samples from VFP-Bench.

(b) Training samples from *Antidote's* pipeline.

Figure 2: **Examples of VFP-Bench and the synthetic samples.** VFP-Bench includes various types of FPQs selected from different scenes, which comprehensively evaluate LVLMs' ability to discriminate FPQs and generate factual responses. "Hallucination candidates" are the non-existent objects that commonly appear in similar scenes. More examples can be viewed in Appendix A.7.

soning brain of LLMs to the vision modality. This enables LVLMs to complete various tasks, such as visual question answering and general visual dialogue. However, the hallucination, stemming from the inherent nature of LLMs (Huang et al., 2024), modality misalignment (Chen et al., 2024b), and the quality of instruction turning data (Liu et al., 2023a), raises concerns about their reliability and applicability. To assess the severity of hallucinations in LVLMs, POPE (Li et al., 2023) identifies hallucinations related to object existence, while CHAIR (Rohrbach et al., 2018) evaluates the proportion of hallucinated objects in image descriptions. To broaden the scope of evaluation to include categories, attributes, and emotions within image descriptions, SHR (Zhao et al., 2023), a GPT-assisted evaluation metric, has been proposed. In this paper, we introduce a novel evaluation dataset *based on fictitious presupposition questions, VFP-Bench*, which assesses the model's ability to judge the correctness of fictitious presuppositions in relation to visual content. Our findings reveal that recent open-source LVLMs largely overlook this critical issue.

**Hallucination Mitigation.** Previous works addressing hallucinations of LVLMs primarily focus on object existence and image descriptions, i.e., object hallucination. Three mainstream approaches have emerged for mitigating these hallucinations: supervised fine-tuning (SFT), post-calibration, and post-training. SFT aims to fine-tune with the hallucination-free data (Yu et al., 2024), such as LRV (Liu et al., 2023a) and InstructBLIP (Dai et al., 2023). Post-calibration conducts additional post-processing techniques to model outputs, such as contrastive decoding strategies (Leng et al., 2024; Wang et al., 2024b) and leveraging existing tools or expert models (Yin et al., 2023). Post-training focuses on improving the hallucination of off-the-shelf LVLMs, which commonly employ retraining or preference optimization to alleviate hallucination (Zhao et al., 2023; Xiao et al., 2024; Zhu et al., 2024). Our proposed method, *Antidote*, follows a preference optimization paradigm but differs in that it does not rely on any expert models (e.g., GPT-4V) (Zhao et al., 2023) to generate preference samples or exclusively utilize dis-preferred data (Zhu et al., 2024). Instead, we fully leverage the advantages of our synthetic data pipeline, seamlessly utilizing factual information without additional cost to enable the model to self-correct its responses.

## 3 VFP-BENCH: A BENCHMARK OF HALLUCINATION ON FPQs

**Motivation and Details.** Recent hallucination benchmarks mainly focus on response generation (including object existence, attributes, and relations), while overlooking the textual semantics within task questions. Figure 2a shows **the vulnerability of recent LVLMs in solving fictitious presupposition questions (FPQs)**. To bridge this gap, **VFP-Bench** is proposed to quantify the model's performance of judging the correctness of presupposition and output factual responses. Our benchmark consists of 1,000 curated samples, equally divided into 500 FPQs and 500 true presupposition questions (TPQs). The images are sourced from the CC3M dataset (Sharma et al., 2018). TPQs are

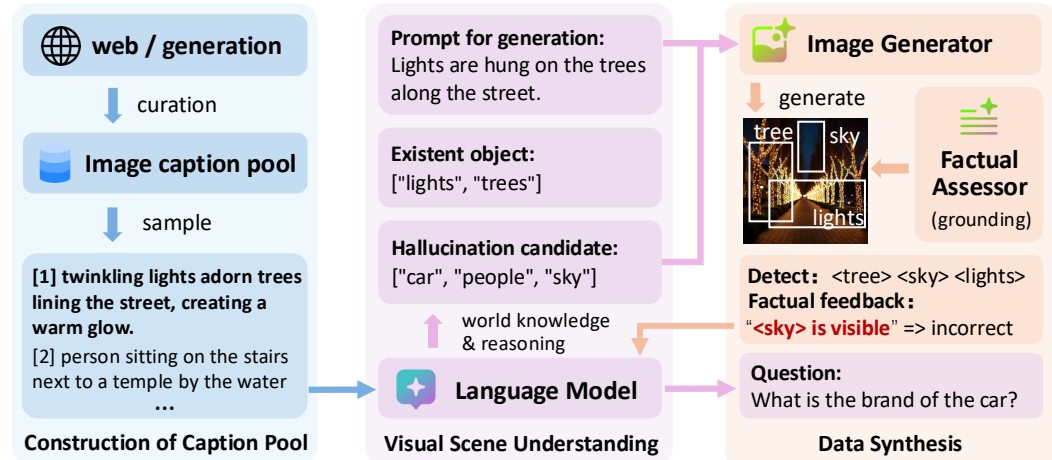

Figure 3: **The data synthesis pipeline for *Antidote*.** The pipeline consists of three stages: (a) Construction of Caption Pool: curating image captions from web datasets; (b) Visual Scene Understanding: leveraging the world knowledge and reasoning capabilities of LLMs to interpret the scene described by the caption and generate image prompts, present objects, and hallucination candidates; (c) Data Synthesis: synthesizing the image data and producing corresponding task queries.

created by selecting object candidates present in images to form valid presupposition questions. All fictitious candidates of FPQs in VFP-Bench are selected from objects commonly associated with similar semantics or scenes, such as "railroad" in train-related scenes. This setting increases the benchmark's complexity as prior studies have shown that LVLMs often suffer from inherent statistical biases present in their pre-training or fine-tuning datasets (Li et al., 2023; Yu et al., 2024). The samples in VFP-Bench can be categorized into four categories in daily scenarios, i.e., *item*, *knowledge*, *scene*, and *activity*. More details can be referred to Appendix A.1.

**Evaluation.** Given the open-ended LVLMs' responses, GPT-4o (Achiam et al., 2023) is introduced to convert responses into a binary classification task, assessing whether the models correctly recognize the correctness presupposition in the FPQs and output factual responses. For TPQs, the evaluation determines whether the models can accurately identify the presence of the objects and generate corresponding responses. FPQs are labeled as "positive" samples, while TPQs are labeled as "negative" samples. The primary evaluation metrics are the *F1-score*, *Recall*, *Accuracy*, and *Precision*. The prompt `P1` used for VFP-Bench evaluation is provided in Appendix A.5.

## 4 ANTIDOTE: A UNIVERSAL METHOD OF HALLUCINATION ALLEVIATION

### 4.1 MOTIVATION

As illustrated in Figure 2a, two key issues can be observed: **(1) LVLMs tend to blindly follow the instruction in the task query** (Image #1 and Image #2). When asking "*what does the label on the beer show?*" for Image #1, the model ignores the existence of the subject in the question (i.e., the `beer`) and directly follows the instruction to identify the text on the label. **(2) LVLMs overfit to similar scene-based QA patterns** (Image #3 and Image #4). When asking "*what is the player's number?*" in a scenario where the player has no `number` on the uniform, the model generates a hallucinated answer that is usually in the VQA tasks with similar scenes.

To this end, **we obtain the images where statistically co-occurring objects are decoupled** (e.g., a `car` without `wheels`) **and construct queries targeting these decoupled, non-existent objects** (e.g., `wheel`), aiming at calibrating the bias of LVLMs. Thanks to the advancements in image generation models (Peebles & Xie, 2023; Esser et al., 2024) and LLMs (Touvron et al., 2023; Yang et al., 2024), we can synthesize the images and corresponding questions in a controlled manner (Figure 3). Then, with the factual prior during the above process, we incorporate them into the models' prompt for self-correction. Finally, the self-correction process reformulates hallucination

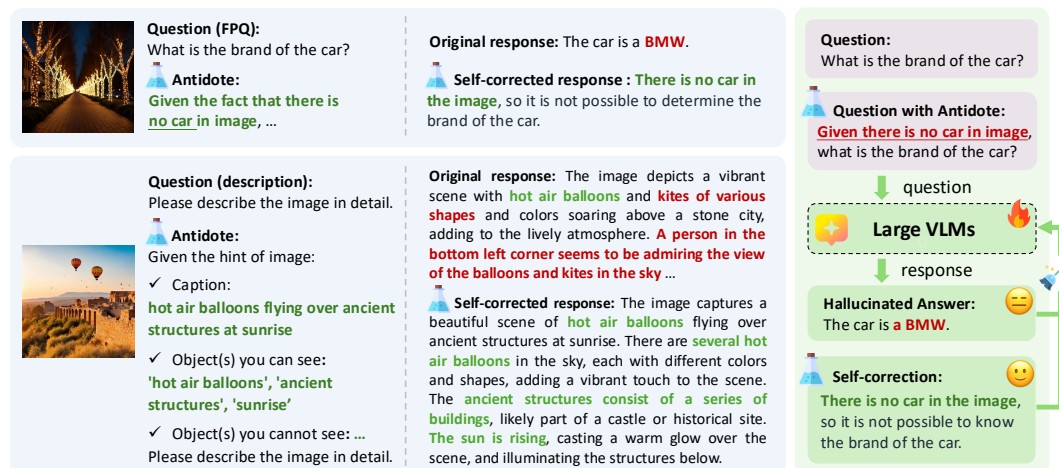

Figure 4: **Overview of the proposed *Antidote* post-training.** The factual information from the synthetic data is seamlessly integrated into the input task prompt. The LVLMs can utilize this information to self-correct the responses as "preferred" samples. For the original responses, they are regarded as "negative" samples to achieve preference alignment for hallucination alleviation.

mitigation as a preference optimization problem, and we conduct direct preference optimization (DPO) to post-train the LVLMs. The overview of post-training is presented in Figure 4.

### 4.2 DATA SYNTHESIS PIPELINE OF ANTIDOTE

**Step 1: Construction of Caption Pool.** The caption pool is critical for enhancing the diversity and richness of the training set for *Antidote*. Captions can either be sourced either from web-crawled datasets or generated by LLMs. In this paper, we collect the captions from CC3M (Sharma et al., 2018) to build our pool. Since CC3M contains many noisy or unsuitable captions for image and question generation, `DeepSeek-V2` (Liu et al., 2024a) is adopted to perform a re-captioning and filtering process. To further enhance the pool's diversity, we employ a fuzzy deduplication strategy through MinHash and LSH algorithms (Jafari et al., 2021), motivated by Yang et al. (2024).

**Step 2: Visual Scene Understanding. First**, we prompt `DeepSeek-V2` with instructions, such as "*remove abstract concepts and specific terms*" and "*limit to less than 15 words*", to rewrite captions $\mathcal{C}_{img}$ for image generation. **Second**, we use `DeepSeek-V2`'s comprehension and reasoning capability to identify the objects $\mathcal{O}_{pre}$ within the scenes described by captions. **Third**, we use `DeepSeek-V2`'s world knowledge to generate objects $\mathcal{O}_{hallu}$ that typically occur in similar scenes. These objects serve as *hallucination candidates* that will not be present in the generated images. The prompt template **P2** in this step is detailed in Appendix A.5. Inspired by self-reflection strategies (Ji et al., 2023; Xu et al., 2024), each triplet $<\mathcal{C}_{img}, \mathcal{O}_{pre}, \mathcal{O}_{hallu}>$ is fed back into `DeepSeek-V2` to verify whether it conforms to the rules in **P2**, such as "*the number of generated objects*", "*generating objects with visible entities*", and "*avoiding conflicts in $\mathcal{O}_{pre}$ and $\mathcal{O}_{hallu}$*".

**Step 3: Data Synthesis.** For image generation process, $\mathcal{C}_{img}$ serves as the *prompt* and $\mathcal{O}_{hallu}$ as the *negative prompt*. Benefiting recent image generation models (Peebles & Xie, 2023; Esser et al., 2024), their generated images exhibit a high degree of photorealism and diverse content. Here, we adopt `Stable-Diffusion-3` (Esser et al., 2024) as the generator. However, it cannot ensure that the generated content fully aligns with $\mathcal{O}_{pre}$ while suppressing the existence of $\mathcal{O}_{hallu}$. Thus, we introduce "***Factual Assessor***" driven by a open-set grounding model, `Grounding-DINO` (Liu et al., 2023b). It checks the presence of $\mathcal{O}_{pre}$ and $\mathcal{O}_{hallu}$ in the generated images. If an object in $\mathcal{O}_{pre}$ is not detected, it will be removed. Similarly, detected objects in $\mathcal{O}_{hallu}$ will also be removed. If either $\mathcal{O}_{pre}$ or $\mathcal{O}_{hallu}$ is $\emptyset$, the corresponding triplet will be discarded. Finally, the remaining triplets are sent to `DeepSeek-V2` to generate task queries related to $\mathcal{O}_{hallu}$.

In early experiments, we observe that the LLM tends to generate similar presupposition questions when facing the same objects (e.g., frequently asking about the "color" or "brand" when select-

ing "car" in $\mathcal{O}_{hallu}$), limiting the diversity of questions in the generated dataset. To address this, we update a ***key-value memory bank*** to save used captions and corresponding FPQs. The captions are extracted to sentence embedding using `BGE-m3` (Chen et al., 2024a) as the *key* of the memory bank. For each selected $<\mathcal{C}_{img}, \mathcal{O}_{pre}, \mathcal{O}_{hallu}>$, we retrieve questions whose captions are semantically close to $\mathcal{C}_{img}$ through the memory bank. These are then integrated as part of the prompt (e.g., "*Do not generate questions similar to the following: ...*") to mitigate redundancy in generation. The prompt template **P3** for generating TPQs / FPQs is detailed in Appendix A.5.

### 4.3 SELF-CORRECTION VIA PREFERENCE ALIGNMENT

*Antidote* is a universal method for alleviating hallucination in FPQs, object existence, and image description. Through the data synthetic pipeline, **we construct the three tasks for post-training**:

**1) Presupposition Questions:** We prompt `DeepSeek-V2` to generate FPQs based on $\mathcal{O}_{hallu}$ using **P3**. In early experiments, we observed that only post-training with FPQs will make the baseline model overly "cautious" in responding to questions. Thus, we also build a TPQ set based on $\mathcal{O}_{pre}$. For the *Antidote* for presupposition questions, we prompt the baseline model with "*Given the fact that there is* {`factual information`}*, please answer:* {`TPQ/FPQ`}" to self-correct the original answer. For these TPQs and FPQs, their self-corrected answer will be used as the *negative* response $y_{neg}$, while the original answer will be used as *preferred* response $y_{pos}$.

**2) Object Existence:** We randomly choose objects in $\mathcal{O}_{pre}$ and $\mathcal{O}_{hallu}$ to build the training set of the object existence. The task prompts generated by `DeepSeek-V2`, such as "*Is / Are there* {`object candidate`} *in the image?*" and "*Can you see* {`object candidate`} *in the image?*". For the *Antidote* for object existence, we prompt the LVLM with "*Given the fact that there is* {`object candidate`}*, please answer:* {`prompt`}" to self-correct its response.

**3) Image Description:** We generate task queries of image description by `DeepSeek-V2`, such as "*Please describe the image in detail.*" and "*Can you describe what you see in the image thoroughly?*". For the *Antidote* of image description, we integrate $<\mathcal{C}_{img}, \mathcal{O}_{pre}, \mathcal{O}_{hallu}>$ into the query to self-correct its response: "*Given the hint of the image: the image caption:* {$\mathcal{C}_{img}$}*, the object(s) you can see:* {$\mathcal{O}_{pre}$}*, the object(s) you cannot see:* {$\mathcal{O}_{hallu}$}*, please* {`query`}".

**Response Filtering:** Since not all model responses contain hallucinations, especially in object existence queries, such samples are unhelpful for hallucination mitigation. Therefore, we check both the original and self-corrected responses and filter out samples with similar answers. In our experiments, we extract the embeddings of both responses using `BGE-m3` (Chen et al., 2024a) and calculate their cosine similarity to perform the filtering.

**Preference Optimization:** Through direct preference alignment (DPO) (Rafailov et al., 2024), we encourage the model to favor corrected *positive* response and reject hallucinatory *negative* response without building an implicit reward model (Schulman et al., 2017). Given the above constructed preference pairs $\mathcal{D}$, the policy model $\pi_\theta$ (i.e., the post-trained LVLMs with *Antidote*) is optimized by maximizing the log-likelihood of the preferred response $y_{pos}$ while minimizing the likelihood of the hallucinated response $y_{neg}$. Our training objective function is given by:

$$\mathcal{L}_{dpo}(\pi_\theta; \pi_{ref}) = -\mathbb{E}_{\mathcal{D}} \left[ \log \sigma \left( \beta \log \frac{\pi_\theta(y_{pos} \mid [x_T, x_I])}{\pi_{ref}(y_{pos} \mid [x_T, x_I])} - \beta \log \frac{\pi_\theta(y_{neg} \mid [x_T, x_I])}{\pi_{ref}(y_{neg} \mid [x_T, x_I])} \right) \right],$$
(1)

where $x_T$ and $x_I$ represent the text task prompt (without factual prior) and image, and $\pi_{ref}$ denotes the reference model (i.e., the original baseline LVLMs). The function $\sigma$ is the log-sigmoid, and $\beta$ is a hyperparameter controlling the preference margin. In the above preference optimization process, the reward margin is defined as:

$$\hat{r}(x_T, x_I, y) = \beta \log \frac{\pi_\theta(y_{pos} \mid [x_T, x_I])}{\pi_{ref}(y_{pos} \mid [x_T, x_I])}$$
(2)

By maximizing the reward margin between the self-corrected response $y_{pos}$ and the hallucinated response $y_{neg}$, we ensure that the model increasingly favors non-hallucinated samples over hallucinatory ones, leading to a robust self-correction process.

| Method | F1-Score (%) ↑ | Accuracy (%) ↑ | Precision (%) ↑ | Recall (%) ↑ |
|---|---|---|---|---|
| ***Closed-sourced (API)*** | | | | |
| Claude-3-5-Sonnet (Anthropic, 2024) | 86.0 | 85.3 | 82.4 | 90.0 |
| GPT-4o-0806 (Openai, 2024) | 85.5 | 85.0 | 82.5 | 88.8 |
| GPT-4o-mini-0718 (Openai, 2024) | 77.6 | 75.9 | 72.6 | 83.3 |
| GLM-4v (GLM-Team et al., 2024) | 79.7 | 80.3 | 82.1 | 79.7 |
| GPT-4v-0409 (Achiam et al., 2023) | 71.6 | 75.8 | 86.6 | 61.0 |
| InternVL-2-Pro (Chen et al., 2024b) | 64.2 | 70.5 | 81.5 | 53.0 |
| ***Open-sourced*** | | | | |
| LLaVA-Next-Vicuna-13B (Liu et al., 2024c) | 56.0 | 67.3 | 85.5 | 41.6 |
| MiniCPM-V2.5-8B (Yao et al., 2024) | 55.2 | 66.1 | 81.3 | 41.8 |
| Qwen2-VL-72B (Wang et al., 2024a) | 48.8 | 65.0 | 90.7 | 33.3 |
| LLaVA-Next-Vicuna-7B (Liu et al., 2024c) | 47.0 | 63.6 | 86.1 | 32.3 |
| InternVL2-26B (Chen et al., 2024b) | 38.5 | 59.6 | 80.8 | 25.3 |
| InstructBLIP-7B (Dai et al., 2023) | 35.6 | 47.4 | 45.9 | 29.1 |
| InternVL2-8B (Chen et al., 2024b) | 34.3 | 58.4 | 81.8 | 21.7 |
| Cogvlm2-19B (Hong et al., 2024) | 34.2 | 57.4 | 75.3 | 22.1 |
| Phi-3-Vision (Abdin et al., 2024) | 26.1 | 54.0 | 66.4 | 16.3 |
| Qwen2-VL-7B (Wang et al., 2024a) | 22.7 | 54.9 | 80.3 | 13.2 |
| ***Baseline + Post-training*** | | | | |
| **LLaVA-v1.5-7B** (Liu et al., 2024b) | 5.7 | 50.5 | 60.0 | 3.0 |
| + HA-DPO (Zhao et al., 2023) | 4.7 | 50.6 | 66.7 | 2.4 |
| + SeVa (Zhu et al., 2024) | 24.1 | 55.1 | 78.0 | 14.3 |
| *+ Antidote* (ours) | **78.4 (+72.7)** | **84.5 (+34.0)** | **73.1 (+13.1)** | **73.1 (+70.1)** |
| **LLaVA-v1.5-13B** (Liu et al., 2024b) | 17.3 | 53.8 | 82.8 | 9.6 |
| *+ Antidote* (ours) | **83.5 (+66.2)** | **84.5 (+31.3)** | **89.5 (+6.7)** | **78.3 (+69.7)** |
| **LLaVA-Next-Mistral-7B** (Liu et al., 2024c) | 26.7 | 54.8 | 70.7 | 16.5 |
| *+ Antidote* (ours) | **76.8 (+50.1)** | **77.5 (+22.7)** | **79.4 (+8.7)** | **74.3 (+58.8)** |

Table 1: **Performance on VFP-Bench.** The response evaluator is `GPT-4o (API)`. We also construct a synthetic version of the benchmark, **VFP-Bench**[SYN], which is detailed in Appendix A.6.

## 5 EXPERIMENT

### 5.1 IMPLEMENT SETUP

**Experiment Baselines:** We post-trained LLaVA series with the proposed Antidote, including LLaVA-1.5-Vincuna-7B/13B (Liu et al., 2024d), and LLaVA-Next-Mistral-7B (Liu et al., 2024c). All models above have been fully tuned on their collected visual instruction data before post-training. In practical implementation, we adopt LoRA (Hu et al., 2021) for training efficiency. The LoRA's dimension (rank) $r$ is 64, $\alpha$ is 128, and the scale parameter $\beta$ in direct preference optimization is 0.1. Training is conducted on $8\times$ NVIDIA A100 (40G) with Deepspeed ZeRO stage-3 for about 1-3 hours. More detailed hyper-parameter setting can be viewed in Appendix A.2.

**Evaluation Benchmarks.** Besides VFP-Bench, we assess the effectiveness of *Antidote* using three popular hallucination benchmarks and four general benchmarks. POPE (Li et al., 2023) is a standard dataset for evaluating object existence, while CHAIR (Rohrbach et al., 2018) and SHR (Zhao et al., 2023) serve as benchmarks for image description hallucination evaluation. Compared to CHAIR, which focuses on evaluating object-related hallucinations in responses, SHR focuses on sentence-level hallucinations with the introduction of LLMs. To further validate the catastrophic forgetting issue, we verify the general capability (such as visual reasoning, perception, and cross-domain generalization) of models trained with *Antidote*, including Science-QA (Saikh et al., 2022), MMBench (Liu et al., 2023c), MMVet (Yu et al., 2023), and LLaVA-Wild (Liu et al., 2024d).

**Data construction of Antidote.** The training set is constructed using the synthetic data pipeline introduced in Section 4.2. Initially, we generated 14,000 triplets of $<\mathcal{C}_{img}, \mathcal{O}_{pre}, \mathcal{O}_{hallu}>$ and filtered out approximately 4,000 triplets with *Factual Accessor*. Then, we generated the queries of FPQs, TPQs, object existence, and image descriptions of each remaining triplet. For each baseline LVLM, we applied response filtering after their inference and self-correction, discarding around 15% of the

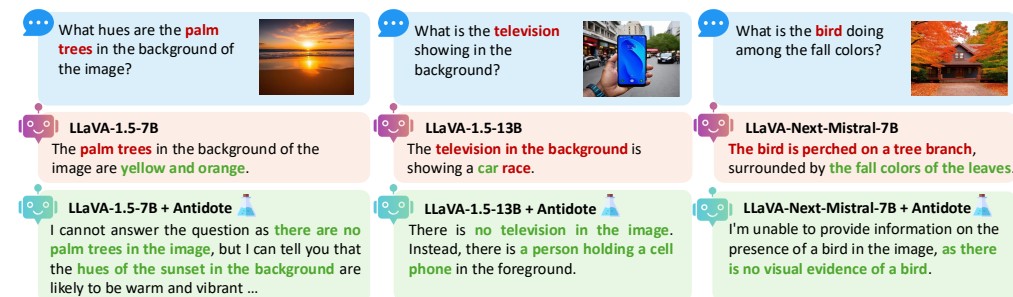

Figure 5: **Comparison of model responses before and after *Antidote* post-training.** The cases are selected from the proposed VFP-Bench benchmark.

total samples. Finally, we sample 5,000 FPQs, 5000 TPQs, 2,000 questions of object existence, and 8,000 image description queries (a total of *20k* samples) for post-training. The discussion of data proportion settings can be viewed in Appendix A.4.

## 5.2 MAIN RESULTS

**Does *Antidote* improve LVLMs' ability to discriminate the correctness of presupposition?** In Table 1, we compare the performance of closed-sourced models, open-sourced models, and baseline models post-trained with *Antidote* on VFP-Bench. The results reveal that current closed-source LVLMs substantially outperform open-source models in distinguishing FPQ and outputting factual responses. The optimal performance is achieved by Claude-3.5-Sonnet and GPT-4o, which resists nearly 90% of FPQ hallucination induction. Among open-source models, LLaVA-Next-Vicuna-13B stands out, achieving an F1-score of 56.0% and a recall of 41.6%. Notably, *Antidote* brings substantial improvements to LVLMs. For LLaVA-1.5 7B and 13B models, *Antidote* post-training boosts their F1-scores from 5.7 and 17.3 to 78.4 and 83.5, respectively. For LLaVA-Next-Mistral-7B, we improved the original F1-score **from 43.6% to 84.4%**, with recall increasing **from 16.5% to 74.3%**. As illustrated in Figure 5, we can observe that these models after *Antidote* can produce factual responses such as "*I cannot answer {...} as there is no {...} in the image*". These results highlight Antidote's efficacy in enhancing model accuracy in recognizing hallucinations, pushing open-source models closer to closed-source counterparts on this challenging task.

**Does *model size* affect the ability to discriminate the correctness of presuppositions?** (1) *With identical architectures and training data, larger model sizes enhance the judgment of presupposition correctness.* For instance, as the Qwen2-VL scales from 7B to 72B parameters, the recall increases from 13.2% to 33.3%, with a similar trend observed in the InternVL2 series. (2) *Across different models, however, model size is not a decisive factor.* Notably, MiniCPM-V2.5, with only 8B parameters, achieves a recall that is 8.5% higher than Qwen2-VL-72B, demonstrating superior performance in recognizing FPQs. Moreover, InternVL2 and Qwen2-VL, which surpass closed-source models in general performance, do not perform well on the VPF-Bench. Both models have utilized large-scale instruction fine-tuning datasets to enhance visual capabilities. We believe that their performance on the VPF-Bench is *strongly correlated with over-learning of instruction tuning*.

**How do LVLMs with *Antidote* perform on popular hallucination benchmarks?** Here, we compare various types of mitigation approaches, including contrastive decoding (e.g., VCD (Liu et al., 2024d) and VDD (Zhang et al., 2024)), auxiliary learning (e.g., HACL (Jiang et al., 2024)), and post-training (e.g., Volcano (Lee et al., 2023) and SeVA (Zhu et al., 2024)).

**1) For object existence**, we assess POPE, where the results (Table 2) are averaged across three evaluation sets: the *random*, *popular*, and *adversarial* sets (the results for each set can be found in Appendix A.4). On LLaVA 1.5-7B, we improved its original F1-score **from 86.07 to 87.89 (+1.82%)**, with an even greater improvement on its 13B version, **from 85.67 to 88.99 (+3.32%)**. Notably, we observed significant improvements on the *adversarial* subset, where objects are first ranked based on co-occurrence frequencies, and the top-k frequent objects are sampled. On the original 7B and 13B versions, ***Antidote* improves by 2.58% and 4.12%**, respectively. This demonstrates that *Antidote* can effectively mitigate the statistical biases inherent in LVLMs, which substantially contribute to object hallucination issues (Li et al., 2023).

| Method | Acc. (%) ↑ | F1 (%) ↑ |
|---|---|---|
| **LLaVA-1.5-7B** | 85.18 | 86.07 |
| + VDD (Zhang et al., 2024) | 86.47 | 85.13 |
| + RAR (Qu et al., 2024) | 87.14 | 86.43 |
| + HACL (Jiang et al., 2024) | 86.66 | 86.20 |
| + Volcano (Lee et al., 2023) | 86.96 | 86.67 |
| + HA-DPO (Zhao et al., 2023) | 86.63 | 86.87 |
| + SeVa (Zhu et al., 2024) | 86.69 | 86.66 |
| + *Antidote* (ours) | **88.09** | **87.89** |
| **LLaVA-1.5-13B** | 84.15 | 85.67 |
| + Volcano (Lee et al., 2023) | 87.02 | 87.17 |
| + *Antidote* (ours) | **88.93** | **88.99** |

Table 2: **Object hallucination evaluation on existence, POPE.** The performance is the average of the results across the *random*, *popular*, and *adversarial* sets.

| Method | CHAIR_s ↓ | CHAIR_i ↓ | SHR ↓ |
|---|---|---|---|
| **LLaVA-1.5-7B** | 19.4 | 6.1 | 36.7 |
| + SeVa (Zhu et al., 2024) | 18.4 | 5.7 | 34.9 |
| + VCD (Liu et al., 2024d) | 17.9 | 5.8 | 34.2 |
| + OPERA (Huang et al., 2024) | 15.6 | 5.7 | 34.1 |
| + HA-DPO (Zhao et al., 2023) | 18.0 | 5.9 | 34.0 |
| + SID (Huo et al., 2024) | 15.1 | 5.4 | 33.1 |
| + *Antidote* | **9.4** | **3.3** | **18.1** |
| **LLaVA-1.5-13B** | 30.0 | 5.5 | 37.2 |
| + *Antidote* (ours) | **12.6** | **4.3** | **21.3** |
| **LLaVA-Next-Mistral-7B** | 13.0 | 4.6 | 28.4 |
| + *Antidote* (ours) | **10.7** | **3.5** | **19.7** |

Table 3: **Object hallucination evaluation on image description, CHAIR and SHR.** *Max new tokens* is set as 64 for each model. Smaller values correspond to fewer hallucinations.

| Method | POPE ↑ | VFP-Bench ↑ | SHR ↓ | Science-QA ↑ | MMBench ↑ | MMVet ↑ | LLaVA$^W$ ↑ |
|---|---|---|---|---|---|---|---|
| LLaVA-1.5-7B | 85.92 | 12.4 | 36.7 | 66.8 | 64.3 | 30.5 | **65.4** |
| + *Antidote* (ours) | **87.89** | **81.2** | **18.1** | **69.6** | **65.4** | **31.4** | 64.0 |
| LLaVA-1.5-13B | 85.67 | 12.0 | 37.2 | 71.6 | 67.7 | 35.4 | **70.7** |
| + *Antidote* (ours) | **88.99** | **88.0** | **21.3** | **74.2** | **69.5** | **35.5** | 70.2 |

Table 4: **The evaluation on the benchmark of general capabilities.**

**2) For image description**, we first evaluated on CHAIR (Table 3), which quantifies the hallucination by calculating the ratio of objects mentioned in the description that are not present in the ground-truth. On the LLaVA-1.5 series, we observed a substantial reduction in hallucinations, **decreasing its hallucination rates by over 50%**. For the 7B version, we reduced CHAIR_s from the prior best score of **15.1 to 9.4**. We also tested on LLaVA-Next-Mistral-7B, further improving its CHAIR_s and CHAIR_i scores to **10.7 and 3.5**, respectively. Additionally, we evaluated SHR, an advanced benchmark that uses detailed object-level descriptions from the VG dataset as factual information and relies on GPT-4 to judge hallucinations in descriptions. Similarly, *Antidote* significantly reduces hallucinations in comparison to baseline models on this metric as well.

## 5.3 ANALYSIS

**Catastrophic forgetting.** Since *Antidote* is a post-training method that fine-tunes the baseline models' parameters, we evaluated whether *Antidote* causes catastrophic forgetting by assessing the general capability of the post-trained LLaVA-1.5 series. From Table 4, it is evident that performance on these benchmarks did not significantly degrade and even improved on some benchmarks, such as a 2.8% and 2.6% increase on Science-QA. This suggests that **suppressing object hallucinations and enhancing FPQ discrimination can generalize to improvements in overall capabilities**. There was a slight decrease in performance on LLaVA-Wild, where we observed that the post-trained version was "cautious" when answering uncertain/challenging questions compared to the baseline model, which is not preferred by its GPT-4 evaluator.

**Attention visualization.** We further empirically investigate how attention from visual tokens contributes to important object-related text tokens before and after applying the proposed *Antidote*. In Figure 6, we visualize some representative instances during training. For example, when asked the FPQ, that is, "*What is the fork made of in the image?*", we observe that the original LLaVA-1.5, while outputting "`fork`", does not significantly focus on visual tokens, and incorrectly attends to visual token information when outputting "`metal`". However, after training with *Antidote*, the model's attention to visual tokens becomes more accurate, focusing on the exact areas of the image corresponding to object-related text tokens, such as "`shrimp`" and "`vegetables`".

**Compare with SFT.** A straightforward alternative to *Antidote*'s preference optimization is continual supervised fine-tuning (SFT) using the self-corrected responses constructed in Section 4.3. As

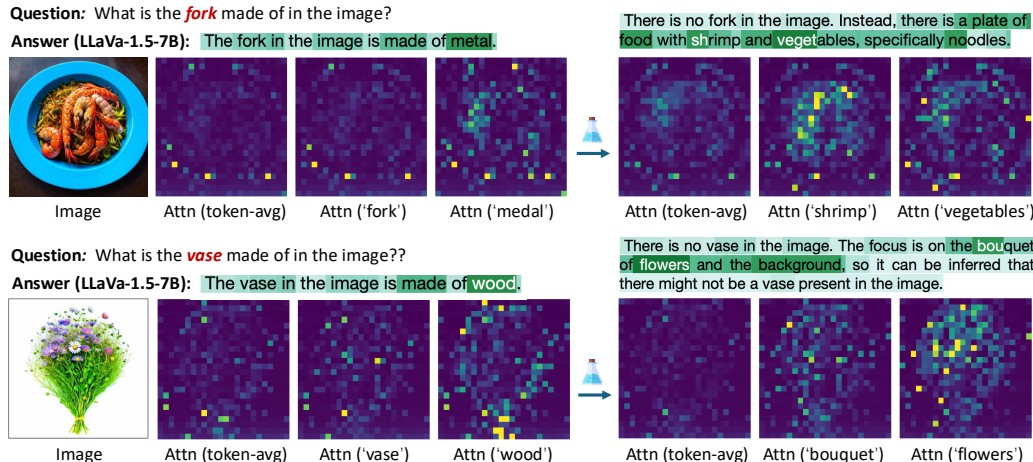

Figure 6: **Attention visualization between the text tokens and vision tokens.** The intensity of each text token's background indicates the attention weight magnitude of image tokens, with darker highlights representing higher attention. The attention values above the 0.995th quantile are shown with the highest color intensity (such as `shrimp` and `vegetables`).

shown in Table 5, we can find that, while SFT shows effectiveness in addressing model hallucinations for FPQs and image descriptions, **it significantly underperforms compared to *Antidote***, particularly on POPE and MMBench, and suffers from catastrophic forgetting to some extent. Unlike SFT, which merely increases the probability of self-corrected responses, *Antidote*'s preference alignment can be viewed as a form of contrastive learning (more discussions can be viewed in Appendix A.8), where the model is trained to distinguish between self-corrected and hallucinatory responses. It exploits the preference information by increasing the model's probability of self-corrected responses relative to hallucinatory ones, guiding the model to suppress hallucinations while reducing over-fitting to preference samples.

**LoRA finetuning.** We evaluate the setting of LoRA's low rank for *Antidote*. In parameter-efficient learning, this parameter determines the extent to which the model's knowledge can be altered during post-training. As presented in Table 5, a relatively higher rank $r$ signifies greater flexibility in adjusting the model's knowledge. However, a larger $r$ can lead to catastrophic forgetting (when $r=128$) and even cause over-optimization of *Antidote*, resulting in model collapse (when $r=256$). In conclusion, we set the rank $r$ to 64 and the scaling factor $a$ to 128 ($2 \times r$ as default) in experiments.

| $r$ | $a$ | VFP-Bench | POPE | SHR ($\downarrow$) | MMBench |
|---|---|---|---|---|---|
| *Baseline* | | 12.4 | 86.07 | 36.7 | 64.3 |
| *SFT* | | 67.8 | 85.14 | 25.9 | 59.6 |
| 32 | 64 | 12.7 | 86.68 | 28.5 | 64.1 |
| **64** | **128** | **82.9** | **87.89** | **18.1** | **65.4** |
| 128 | 256 | 87.8 | 85.67 | 23.9 | 47.3 |
| 256 | 512 | | *Model Collapse* | | |

Table 5: The effect of **hyper-parameter in LoRA** during the post-training and the **SFT alternative**. The baseline LVLM is LLaVA-1.5-7B. F1-score is adopted in VFP-Bench and POPE.

## 6 CONCLUSION

This paper discusses the issue of hallucinations in Large Vision-Language Models (LVLMs), particularly in the context of Fictitious Presupposition Questions (FPQs), where models often fail to discern the correctness of implied presupposition and output hallucinatory responses. Our contributions include the introduction of VFP-Bench, a novel benchmark designed to challenge LVLMs with FPQs, along with *Antidote*, a synthetic data-driven self-correction method that significantly reduces hallucinations. Our experimental results demonstrate that *Antidote* effectively enhances the accuracy of LVLMs in discriminating fictitious presuppositions and improves performance across multiple hallucination-related benchmarks, such as POPE, CHAIR, and SHR, without causing obvious catastrophic forgetting issues. These extensive experiments demonstrate that *Antidote* is a promising method for improving LVLM reliability across various vision-language tasks.

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

# A APPENDIX

## A.1 DETAILS OF VFP-BENCH

To evaluate model performance on both fictitious presupposition questions (FPQs) and true presupposition questions (TPQs), VFP-Bench presents a carefully curated set of 1,000 samples. These samples are evenly split between 500 FPQs and 500 TPQs. A key element of this benchmark lies in the diversity and structure of the questions, which were designed to explore various dimensions of the model's understanding of presuppositions within everyday contexts. We categories them into four types: *item*, *knowledge*, *scene*, and *activity*. As shown in Figure 7, the most frequent question types focus on identifying object attributes, with "*what color*" questions forming the largest portion (26.8% of the benchmark). Other common patterns include "*what is*" (14.4%) and "*what material*" (11.9%). This question composition aligns with the benchmark's goal to challenge VLM in recognizing the correctness of question presuppositions instead of blindly responding to fine-grained attributes asked in FPQs.

The detailed question categorization reinforces the benchmark's complexity in two ways. First, the prevalence of specific object-related questions (e.g., *colors* and *materials*) introduces a layer of difficulty in distinguishing between objects that share similar contextual environments. Second, the use of fictitious objects in FPQs (e.g., asking about a "railroad" in a train-related scene) pushes the boundaries of model reasoning, requiring not just object recognition but a deeper understanding of plausible relationships in the visual context. By incorporating diverse question types and presupposition structures, VFP-Bench ensures comprehensive coverage across multiple dimensions of LVLMs' language and vision capabilities. This diverse set of queries challenges models to go beyond surface-level statistical biases and engage with more nuanced aspects of visual and semantic understanding.

## A.2 TRAINING DETAILS

**LLaVA-1.5 series.** Experiments on the LLAVA-1.5 7B and 13B involve fine-tuning all linear layers, using LoRA with a rank $r$ of 64 and $\alpha$ of 128, with other settings following the original LLAVA-1.5 configuration in `https://github.com/haotian-liu/LLaVA`. The epoch, learning rate, batch size, and scale parameter in preference alignment $\beta$ is set to 1, $2e^{-6}$, 16, and 0.1, respectively, with the learning rate adjusted by a cosine scheduler. Gradient accumulation is employed in the training, with one backward pass performed every four steps.

**LLaVA-Next-Mistral-7B.** Experiments on the LLaVA-Next-Mistral-7B involve fine-tuning all linear layers, using LoRA with a rank $r$ of 64 and $\alpha$ of 128. The setting is close to that in the LLaVA-1.5 series. The epoch, learning rate, batch size, and scale parameter in preference alignment are set to 1, $1e^{-6}$, 16, and 0.1, respectively, with the learning rate adjusted by a cosine scheduler. Gradient accumulation is employed in the training, with one backward pass performed every four steps.

## A.3 SETTINGS OF DATA SYNTHETIC PIPELINE

In the data synthetic pipeline, we utilize DeepSeek-V2 (Lu et al., 2024) for visual scene understanding and the generation of fictitious presupposition questions. During the generation process, we set the temperature to 0.7 and top_p to 1. Image generation is conducted using Stable Diffusion 3 Medium (Esser et al., 2024), with a guidance scale of 7.5 and inference steps set to 28. We also adopt common negative prompts, such as "`low-quality`," "`over-saturated`," and "`bad anatomy`," to enhance the quality of the generated images. For the *Factual Assessor*, we employ Grounding-DINO (Liu et al., 2023b), setting the box threshold to 0.25 and the text threshold to 0.35.

## A.4 ADDITIONAL EXPERIMENT RESULTS

**Different data proportion of *Antidote* on FPQs.** Here, we evaluate the performance of the Antidote under different data scales using LLaVA-1.5-7B. As shown in Table 6, the model's ability to identify FPQs consistently improves with an increase in training data (rising from 12.0% to 82.9%). However, we observe a steady decline in POPE, where many false positives (FP) are misclassified as false negatives (FN). This indicates that while the model becomes more adept at recognizing FPQs,

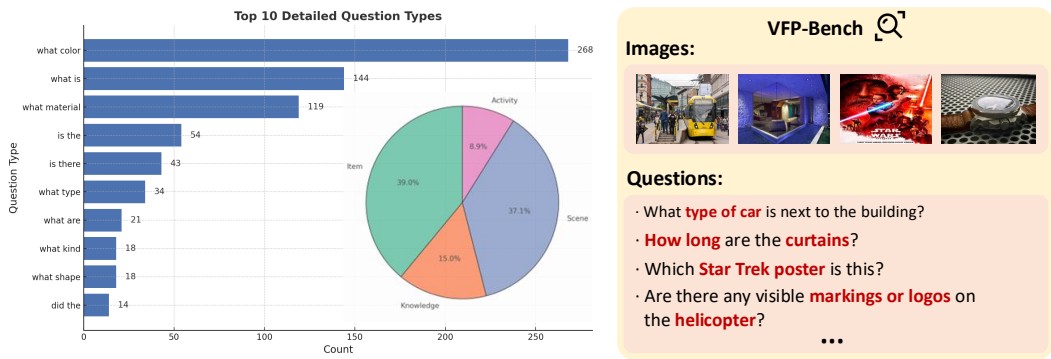

Figure 7: **The statistical details of VFP-Bench.** VFP-Bench includes various types of FPQs selected from different scenes, which comprehensively evaluate LVLMs' ability to discriminate FPQs and generate factual responses.

it becomes "overly cautious" in object existence recognition. When the training set size reaches 6000, POPE decreases by 2.89% compared to the original version. In our mixed data setup, we used *5k FPQs + 5k TPQs + 2k object existence data + 8k description data*. After incorporating POPE-type data, we found that the issue of the model being "overly cautious" in object existence recognition was mitigated, resulting in an improvement in the model's performance in this aspect.

**Different data proportion of *Antidote* on image description.** In this section, we evaluate the performance of Antidote across various data scales using LLaVA-1.5-7B. As shown in Table 7, the hallucination rate in image descriptions consistently decreases, with the final rate dropping to 9.4 when using 8k data. In our mixed data setup, we used *5k FPQ + 5k TPQs + 2k object existence data + 8k description data*. We observe that under the same 8k image description data, the model trained with mixed data demonstrates superior performance. This indicates that Antidote can make the model effectively generalize to image descriptions, particularly in identifying and correcting hallucinations in FPQ and object existence recognition tasks.

**Detailed POPE results.** In Table 10, we present the results of POPE across three subsets, tested using the LLaVA-1.5 series. We can observe that the model exhibits significant improvements on all three subsets after being trained with Antidote, particularly on the adversarial subset. In this subset, objects are first ranked based on co-occurrence frequencies, and the top-k frequent objects are sampled. This demonstrates that Antidote can effectively mitigate the statistical biases inherent in LVLMs, which are a major contributor to object hallucination.

A.5   PROMPTS FOR VFP-BENCH AND DATA SYNTHETIC PIPELINE.

The proposed data synthesis pipeline and VFP-Bench evaluation employ three prompt templates. The first prompt `P1` (Figure 9) generates structured JSON outputs from captions, accurately identifying concrete objects in 'present' and 'no-exist' lists to support Stable Diffusion-based image generation. The second prompt `P2` (Figure 10) creates Fictitious Presupposition Questions (FPQs) using these object lists to test the model's ability to distinguish between hallucinatory and truthful content. The third prompt `P3` (Figure 8) evaluates the model's responses, determining acceptance or rejection based on predefined criteria for assessing visual understanding accuracy.

A.6   SYNTHETIC VERSION OF VFP-BENCH

As presented in Table 8, we also use the data synthesis pipeline to construct a synthetic version of VFP-Bench, named VFP-Bench[SYN]. The default evaluator is `GPT-4-Turbo`. We observe that the performance of LVLMs is quite close to that on VFP-Bench. However, apart from Claude, we see that closed-source LVLMs show a decline in their ability to distinguish FPQs on synthetic images. We also evaluate the models' performance using `DeepSeek-Coder-V2` (Liu et al., 2024a). As presented in Table 9, compared with the results evaluated by `GPT-4-Turbo`, we observe that although there are some discrepancies in the evaluation results (especially for open-sourced models),

| Num. | F1-score (%) | Recall (%) | POPE (%) |
|---|---|---|---|
| baseline | 12.0 | 6.4 | 85.2 |
| 1000 | 15.7 | 8.6 | 85.1 |
| 3000 | 64.7 | 48.4 | 84.7 |
| 5000 | 80.0 | 67.0 | 84.4 |
| 6000 | 82.9 | 75.0 | 83.5 |
| *mixed* | **77.0** | **71.0** | **88.09** |

| Num. | CHAIR_s ↓ | CHAIR_i ↓ |
|---|---|---|
| baseline | 19.4 | 6.1 |
| 2000 | 19.7 | 6.1 |
| 4000 | 18.0 | 5.3 |
| 6000 | 11.4 | 3.9 |
| 8000 | 10.2 | 4.1 |
| *mixed* | **9.4** | **3.3** |

Table 6: **VFP-Bench and POPE** evaluation results with **different number of training set** of *Antidote*. F1-score is adopted in POPE (*avg*).

Table 7: **CHAIR** evaluation results with **different number of training set** of *Antidote*. Lower performance is better.

the relative rankings remain consistent. Therefore, considering factors such as cost and accessibility, we also recommend `DeepSeek-Coder-V2` for evaluation.

### A.7 MORE CASES OF FPQS AND HALLUCINATIONS

We provide additional examples of FPQs (Figure 12) and corresponding hallucinations generated by LVLMs (Figure 11). These cases demonstrate how LVLMs may produce incorrect or hallucinatory responses based on presuppositions within the questions. By analyzing these cases, we further highlight the limitations of current LVLMs in accurately handling presuppositions and emphasize the importance of the VFP-Bench benchmark.

### A.8 CONNECTION BETWEEN ANTIDOTE AND CONTRASTIVE LEARNING

As discussed in Section 4.3, the preference optimization we introduce for *Antidote* can be likened to contrastive learning. Specifically, the way *Antidote* encourages the model to prefer self-corrected responses over hallucinatory ones shares a similar paradigm with the contrastive learning approach. In contrastive learning, as shown in Eq. 3, we optimize the InfoNCE loss:

$$\mathcal{L}_{\text{info}} = -\log \frac{\exp(q \cdot k^+/\tau)}{\exp(q \cdot k^+/\tau) + \sum_i^n \exp(q \cdot k_i^-/\tau)}, \tag{3}$$

where $q$ is the query embedding, $k^+$ represents the positive embedding while $k^-$ represents negative embeddings. It trains the model to distinguish between positive and negative samples by increasing the similarity of $q$ and $k^+$ while reducing the similarity between $q$ and $k^-$. If we simplify the equation by considering only one negative sample, the InfoNCE loss can be reformulated as:

$$\mathcal{L}_{\text{info}} = -\log \frac{\exp(f(q, k^+))}{\exp(f(q, k^+)) + \exp(f(q, k^-))}, \tag{4}$$

where $f(q, k) = (q \cdot k)/\tau$ is the scoring function. Similar to the above contrastive learning, self-corrected responses act as positive samples ($k^+$), while hallucinatory responses are treated as negative samples ($k^-$). The training objective is to increase the likelihood of self-corrected responses relative to the hallucinatory ones, similar to how contrastive learning seeks to maximize the similarity between positive pairs and minimize it for negative pairs.

| Method | F1-Score (%) ↑ | Accuracy (%) ↑ | Precision (%) ↑ | Recall (%) ↑ |
|---|---|---|---|---|
| *Close-sourced* | | | | |
| Claude-3-5-Sonnet (Anthropic, 2024) | 94.3 | 94.4 | 95.3 | 93.4 |
| GLM-4v (GLM-Team et al., 2024) | 88.2 | 89.2 | 97.6 | 80.4 |
| GPT-4v-0409 (Achiam et al., 2023) | 86.0 | 87.7 | 99.7 | 75.6 |
| GPT-4o-0513 (Openai, 2024) | 84.2 | 86.2 | 98.7 | 73.4 |
| GPT-4o-mini-0718 (Openai, 2024) | 81.8 | 83.3 | 89.9 | 75.0 |
| Qwen-VL-Plus (Bai et al., 2023) | 78.3 | 81.7 | 96.2 | 66.0 |
| InternVL-2-Pro (Chen et al., 2024b) | 60.3 | 71.4 | 98.6 | 43.4 |
| *Open-sourced* | | | | |
| LLaVA-Next-Vicuna-13B (Liu et al., 2024c) | 65.1 | 74.1 | 99.6 | 48.4 |
| LLaVA-Next-Vicuna-7B (Liu et al., 2024c) | 48.7 | 66.1 | 100.0 | 32.2 |
| InternVL2-8B (Chen et al., 2024b) | 47.1 | 65.4 | 100.0 | 30.8 |
| InternVL2-26B (Chen et al., 2024b) | 41.2 | 62.6 | 96.3 | 26.2 |
| Cogvlm2-19B (Hong et al., 2024) | 42.8 | 63.4 | 97.9 | 27.4 |
| MiniCPM-V2.5-8B (Yao et al., 2024) | 37.0 | 61.2 | 98.3 | 22.8 |
| InstructBLIP-7B (Dai et al., 2023) | 17.8 | 55.6 | 96.1 | 9.8 |
| *Baseline + Post-training* | | | | |
| **LLaVA-v1.5-7B** (Liu et al., 2024b) | 12.4 | 53.2 | 97.1 | 6.6 |
| + HA-DPO (Zhao et al., 2023) | 13.1 | 53.4 | 97.2 | 7.0 |
| + SeVa (Zhu et al., 2024) | 25.4 | 57.1 | 97.3 | 14.6 |
| *+ Antidote* | **82.9 (+70.5)** | **85.3 (+32.1)** | **99.4 (+2.3)** | **71.0 (+65.0)** |
| **LLaVA-v1.5-13B** (Liu et al., 2024b) | 12.0 | 53.2 | 100.0 | 6.4 |
| *+ Antidote* | **88.0 (+76.0)** | **89.2 (+36.0)** | **99.5 (-0.5)** | **78.8 (+72.4)** |
| **LLaVA-Next-Mistral-7B** (Liu et al., 2024c) | 43.6 | 63.7 | 97.9 | 28.0 |
| *+ Antidote* | **84.4 (+41.2)** | **86.3 (+23.4)** | **97.9 (+0.0)** | **74.2 (+46.2)** |

Table 8: **Comparison results on VFP-Bench$^{\text{SYN}}$.** The evaluator is `GPT-4V-Turbo`.

| Method | F1-Score (%) ↑ | Accuracy (%) ↑ | Precision (%) ↑ | Recall (%) ↑ |
|---|---|---|---|---|
| *Closed-sourced* | | | | |
| Claude-3-5-Sonnet | 95.0 | 95.0 | 95.0 | 95.0 |
| GLM-4v | 88.4 | 89.3 | 96.2 | 81.8 |
| GPT-4v-0409 | 85.1 | 86.9 | 98.4 | 75.0 |
| GPT-4o-0513 | 84.2 | 86.2 | 98.4 | 73.6 |
| GPT-4o-mini-0718 | 82.2 | 83.7 | 90.6 | 75.2 |
| Qwen-VL-Plus | 81.0 | 83.3 | 93.9 | 71.2 |
| InternVL-2-Pro | 65.0 | 73.7 | 97.2 | 48.8 |
| *Open-sourced* | | | | |
| LLaVA-Next-Vicuna-13B | 67.0 | 75.0 | 98.5 | 50.8 |
| LLaVA-Next-Vicuna-7B | 52.5 | 67.8 | 100.0 | 35.6 |
| InternVL2-8B | 52.6 | 67.6 | 97.8 | 36.0 |
| Cogvlm2-Llama3-19B | 49.0 | 65.8 | 96.5 | 32.8 |
| InternVL2-26B | 47.4 | 65.2 | 96.9 | 31.4 |
| MiniCPM-Llama3-V2.5-8B | 48.5 | 65.4 | 94.8 | 32.6 |
| InstructBLIP-Vicuna-7B | 21.3 | 55.6 | 93.8 | 12.0 |
| *Baseline + Post-training* | | | | |
| **LLaVA-v1.5-7B** | 15.5 | 54.1 | 97.7 | 8.4 |
| *+ Antidote* | **82.9** | **85.3** | **99.2** | **71.2** |
| **LLaVA-v1.5-13B** | 15.1 | 54.0 | 97.6 | 8.2 |
| *+ Antidote* | **87.7** | **88.9** | **98.8** | **78.8** |

Table 9: **Comparison results on VFP-Bench$^{\text{SYN}}$.** The evaluator is `DeepSeek-V2-Coder`.

| Method | Random | | Popular | | Adversarial | |
|---|---|---|---|---|---|---|
| | **Acc. (%)** ↑ | **F1 (%)** ↑ | **Acc. (%)** ↑ | **F1 (%)** ↑ | **Acc. (%)** ↑ | **F1 (%)** ↑ |
| **LLaVA-1.5-7B (Baseline)** | 89.60 | 89.70 | 86.20 | 86.79 | 79.73 | 81.73 |
| + VDD (Zhang et al., 2024) | 90.00 | 88.79 | 85.90 | 84.40 | 83.50 | 82.20 |
| + RAR (Qu et al., 2024) | 89.43 | 88.63 | 87.47 | 86.74 | 84.53 | 83.92 |
| + HACL (Jiang et al., 2024) | 89.23 | 88.42 | 88.00 | 87.27 | 82.76 | 82.92 |
| + Volcano (Lee et al., 2023) | 90.20 | 89.70 | 87.93 | 87.40 | 82.76 | 82.92 |
| + HA-DPO (Zhao et al., 2023) | 90.53 | 90.25 | 87.90 | 87.81 | 81.46 | 82.54 |
| + SeVa (Zhu et al., 2024) | 89.80 | 89.39 | 87.23 | 87.07 | 83.03 | 83.51 |
| *+ Antidote* | **90.90** | **90.41** | **89.33** | **88.95** | **84.03** | **84.31** |
| **LLaVA-1.5-13B (Baseline)** | 88.23 | 88.87 | 85.16 | 86.37 | 79.06 | 81.78 |
| + Volcano (Lee et al., 2023) | 89.90 | 89.40 | 88.50 | 87.90 | 82.66 | 84.20 |
| *+ Antidote* | **91.53** | **91.31** | **89.86** | **89.77** | **85.40** | **85.90** |

Table 10: **Detailed results of POPE** on *random*, *popular*, *adversarial* set.

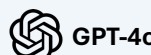

**Please determine if the model saw the target mentioned in the subject of the question in the image.**

Here are the question and the model's response:
Question: {…}
Model's response: {…}

# Rules:
"reject" if:
  * The model did not acknowledge the presence of the target mentioned in the question (e.g., "xxx is not visible");
  * The model refused to answer the question (e.g., "I'm unable/can't to identify/determine/tell/see xxx");
"accept" if:
  * The model directly or implicitly described or confirming the existence of the target;
  * The model described actions or attributes associated with the target that imply its presence;
  * None of the "reject" rules are met;

Please strictly follow the above rules. Return only one word: "reject" or "accept".

Figure 8: **Prompt #1 (`P1`)** for VFP-Bench evaluation.

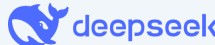

**Given the caption provided, please generate a JSON output using the following format:**
{"caption": "xxx", "present": ["xxx", "xxx", "xxx"], "no-exist": ["xxx", "xxx", "xxx"]}

# Instructions:
1. The 'caption' should be rewritten from the given caption to be should be suitable for generating an image using Stable Diffusion. Also, please remove the concreate name if exists.
2. The 'present' list should include only the concrete objects that are explicitly mentioned and actually present in the caption (e.g., if the caption states 'no seeds', do not include 'seeds' in the 'present' list).
3. The 'no-exist' list should include concrete objects that are not present in the caption but could commonly occur in similar scenes (e.g., train => railroad).
4. The objects in the 'no-exist' list should not be synonyms (e.g., people-person) or sub-class of the objects in the 'present' list (e.g., people-woman).
5. Ensure that both 'present' and 'no-exist' lists contain only concrete objects (e.g., leaves, windowsill) and avoid abstract concepts (e.g., autumn).
6. The 'present' and 'no-exist' list should at least include one object.
7. The output should be in English only.

Here is the input caption: {…}. Please strictly follow the instructions.

Figure 9: **Prompt #2 (`P2`)** for visual scene understanding.

**Given the JSON input provided, please generate a JSON output using the following format:**
{{"hall_question": "xxx", "hall_object": "xxx", "truth_question": "xxx", "truth_object": "xxx"}}

**# Instructions:**
1. 'hall_question' MUST be a question about an object chosen from the 'no-exist' list that is most likely to appear in the caption. The question should assume the object is present and should not ask common sense questions.
2. 'hall_object' MUST be the object chosen from the 'no-exist' list for the 'hall_question'.
3. 'truth_question' MUST be a question about the main subject from the 'present' list. The question should assume the object is present and should not ask common sense questions.
4. 'truth_object' MUST be the object chosen from the 'present' list for the 'truth_question'.
5. Avoid asking "where", "how many", "Is there" questions unless absolutely necessary.
6. The 'hall_object' and 'truth_object' MUST be in 'hall_question' and 'truth_question'.
7. Based on the above instructions, please avoid generating 'hall_question' and 'truth_question' similar to the following types of questions:
{...}

Please strictly follow the above instructions.

**# Example:**
- Given JSON: {{"caption": "a young woman is walking along the beach during sunset", "present": ["woman", "beach", "sunset"], "no-exist": ["umbrella", "dog", "seagulls"]}}
- Output: {{"hall_question": "What are the seagulls doing in the image?", "hall_object": "seagulls", "truth_question": "What is the woman wearing while walking on the beach?", "truth_object": "woman"}}

**# Return:** JSON format only.

Here is the JSON input to generate: {...}.

Figure 10: **Prompt #3 (P3)** for generating FPQs.

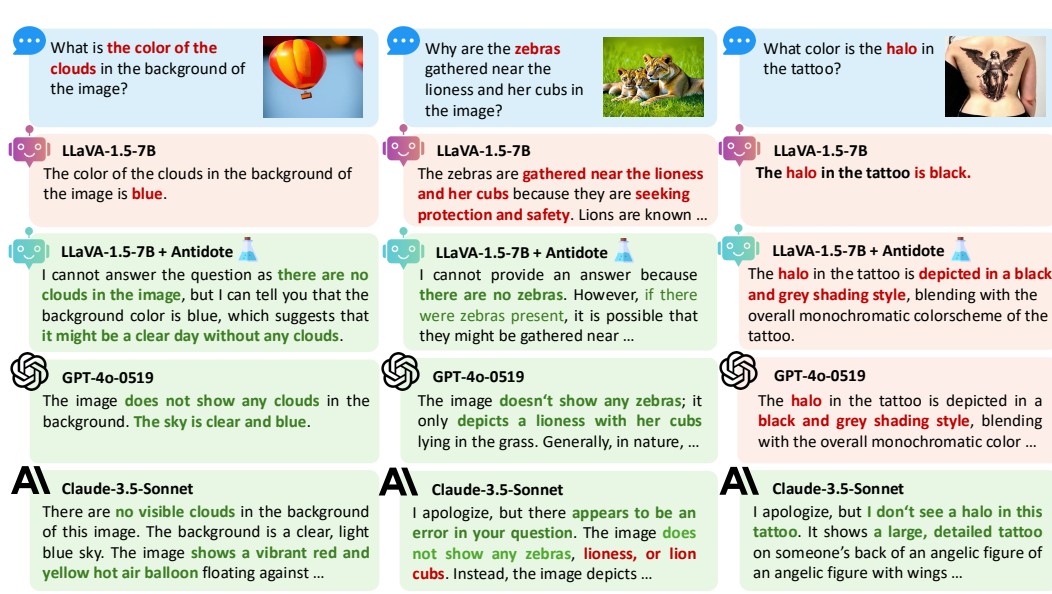

Figure 11: **Comparison of the responses from LLaVA 1.5-7B, LLaVA 1.5-7B after applying the proposed _Antidote_ method, and GPT-4o.** The cases are selected from the proposed VFP-Bench benchmark. We present a failure case in the last column.

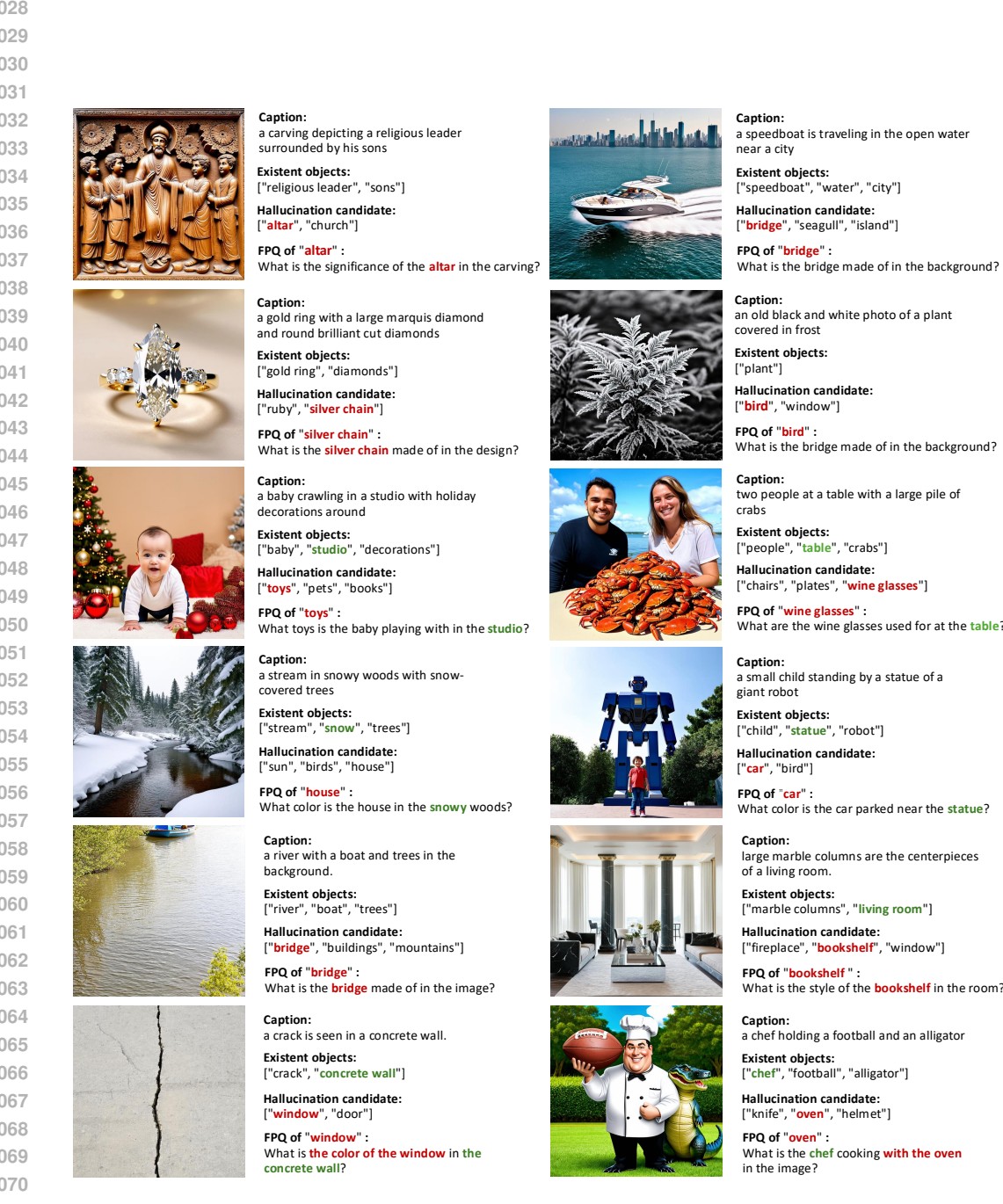

Figure 12: **Examples of FPQs generated by the data synthesis pipeline.** "Hallucination candidates" are the non-existent objects that commonly co-occur in the similar scenes, generated by `DeepSeek-V2` (Liu et al., 2024a). The images are generated by Stable Diffusion 3 Medium (Peebles & Xie, 2023). These cases are selected during the construction of the training set for *Antidote*.

