# OpenReview forum: "Hallucination in LVLMs: Fictitious Presupposition Questions, Benchmark, and Solution"
_ICLR.cc/2025/Conference — ICLR 2025 Conference Withdrawn Submission_

### Official Review · Reviewer_3M22 · 2024-10-29

**Soundness:** 2
**Presentation:** 3
**Contribution:** 3
**Rating:** 5
**Confidence:** 3

**Summary:**

This paper introduces a novel hallucination issue in LVLMs, termed fictitious presupposition questions (FPQs). Compared to the previous hallucination issues in LVLMs, which often focus the binary visual question answering, this study proposes a benchmark that emphasizing long-formed open-ended answers. Additionally, it proposes Antidote, a self-correction method designed to mitigate hallucinations in FPQs. This method is demonstrated its effectiveness on several benchmarks, including POPE, VFP-BENCH, MMBench, using the LLaVA-Next-7B and LLaVA-1.5-7B backbone.

**Strengths:**

- The paper is well-written and easy to follow.
- The introduction of hallucination through fictitious presupposition questions is insightful and novel.
- The proposed mitigation method effectively addresses hallucinations on existing hallucination benchmarks and simultaneously increase general ImageQA benchmarks using the LLaVA-Next-7B and LLaVA-1.5-7B models as backbones.

**Weaknesses:**

[Major]
- The hallucination mitigation method (i.e., Antidote) utilizes the original responses of model as negative examples and the self-corrected responses as positive examples. This approach seems reasonable when the model's original response accuracy is below 50% (as in the case of LLaVA-Next-7B and LLaVA-1.5-7B). However, it may be less effective if the model’s initial responses are already sufficiently accurate (i.e., >= 50% correct).
- Due to the this concern mentioned above, it would be helpful to see experimental results of “MiniCPM-v2.5-8B + Antidote” on benchmarks such as POPE, VFP-Bench, and MMBench (or MM-Vet).

[Minor]
- This paper uses "car" and "wheels" as an example of co-occurring objects, though this example does not appear in the whole paper.
- It appears that co-occurring objects (i.e., hallucination candidates) are generated directly by prompting language model. How can it be demonstrated statistically that these hallucination candidates ("car","people","sky") have a high co-occurrence rate with the existing objects (e.g., trees and lights)?

**Questions:**

Please answer my questions in “weaknesses” section.

---

### Official Review · Reviewer_XnDd · 2024-11-01

**Soundness:** 3
**Presentation:** 3
**Contribution:** 3
**Rating:** 6
**Confidence:** 4

**Summary:**

This paper focusses on hallucinations in large vision language models/multimodal large language models, in the case of an example image and a text question/prompt. Specifically, this paper develops benchmarks and methods which address "fictitious presupposition questions", i.e. questions/prompts to LVLMs which assume the existence of some concept (for example asking an LVLM "What colour is the car?" _presupposes_ that a car is visible in the image, which may not always be the case). The authors show that many open-source models which perform well on a variety of benchmarks suffer from the hallucinations in the fictitious presupposition case. The authors create a benchmark (VFP-Bench) to establish this. Next the authors propose "Antidote", a method to generate preference data which can alleviate hallucinations by creating paired data, one negative example in which the hallucination is present and a positive example which correctly rejects the presupposition of a given prompt—this is done by injecting additional information in the data generation stage which enables a given model to easily reject the presupposition (e.g. "Given that there is no car in this image, what colour is the car?"). This preference data is used to train existing models from the LLaVA series via DPO. Finally, the authors show that training with their preference data improves greatly performance on their proposed benchmark (VFP-Bench) and yields small improvements across a number of other benchmarks (MMBench, MMVet, POPE etc.).

The first main contribution

**Strengths:**

This paper addresses a very important aspect of LVLMs, i.e. they often take the prompts given to them as fact and can be easily led down a path of incorrect assumptions despite it being seemingly obvious from a given image that such assumptions are wrong.

The explanations of the paper are generally very good. Antidote is explained well in Section 4 and the authors have compared to many existing models in the experiments section. The method of post-training, LoRa vs full fine-tuning is an important ablation included in the paper. Finally the experiments which demonstrate that using Antidote leads to the same or slightly improved performance are important given any solution for one problem should at least maintain performance elsewhere to be considered effective.

**Weaknesses:**

I have three main concerns with this paper, which should be addressable during the discussion phase.

1. I find the detail in main paper of VFP-Bench extremely thin. In a 10 page paper, the benchmark on which the main method "Antidote" tries to solve covers approximate 1/3 of a page. The additional details in the appendix (which many readers will not read), largely repeats the details from the main paper. The paper details what the benchmark contains, but it is not clear how the questions and images are curated on a practical level. "TPQs are created by selecting object candidates present in the images to form valid presupposition questions", given that the images are sourced from CC3M, how are the object candidates selected? How are the questions created? How are commonly associated objects chosen for VFP-Bench, is it the same as Antidote?

2. I think there are some similar works which need to be compared to. For example LRV (referenced on L143) suggests paired data (not to be used in a DPO fashion but SFT) to counteract issues very similar to the ones identified in this paper (see Figure 2 of https://arxiv.org/pdf/2306.14565). Really, these similar prior works should be compared to in the paper experiments such as in Table 1.

3. LVLM base models. The authors have applied their Antidote method to LLaVA models only which largely use the same training data (see first Table of https://llava-vl.github.io/blog/2024-01-30-llava-next/). It would be preferable to apply antidote to other models such as InstructBLIP, Qwen-VL etc. This should be doable given the post-training method proposed takes 1-3 hours (L363).

**Questions:**

1. Please provide clarifications on how the proposed VFP-Bench is constructed and if it differs from the pipeline in Antidote? (weakness 1)

2, It would be good if results on VFP-Bench when using similar datasets such as LRV are presented and if the authors omitted such comparisons, to given reasons why. (weakness 2)

3. Do the improvements observed on LLaVA models generalise to other open-source models like Qwen-VL and InstructBLIP etc, if these comparisons were deliberately omitted, please give reasons why.

The authors should check the labelling of Prompt 1, 2, 3 and reference made in the main paper and appendix, I believe there are some typos/mixed references.

I am not experienced in _using_ DPO, but I understand the concept, are the labels for negative and positive responses on L287-L288 correct? I think they should be reversed.

---

### Official Review · Reviewer_SJou · 2024-11-04

**Soundness:** 2
**Presentation:** 3
**Contribution:** 1
**Rating:** 3
**Confidence:** 4

**Summary:**

The hallucination problem of LVLM refers to the phenomenon where the text generated by the model does not align with the content of the image. Previous works focus on mitigating hallucinations in model response generation. The authors highlight that a misleading question may lead to more serious hallucination issue in LVLMs.
- They name the misleading questions as fictitious presupposition questions (FPQs), where a user asks a question assuming the presuppositions of non-existent objects and misleads LVLMs.
- The authors introduce a new benchmark VFP-Bench, to evaluate LVLM’s ability to identify FPQs and generate factual reponses.

Additionally, the authors introduce Antidote to alleviate hallucination in FPQs.
- They synthesize images and generate FPQs by using LLM and image generative models.
- Then, the synthetic FPQ data are fed to LVLMs and the models are improved by using preference optimization. The original responses from models are “negative” while the self-corrected responses are “preferred”.

Antidote significantly improves LLaVA series, by reducing hallucination on VFP-Bench, POPE, CHAIR, and SHR, while maintaining model performance on general LVLM benchmarks.

**Strengths:**

- The figures are clear and helpful.
- Experiments cover different benchmarks and consider the catastrophic
forgetting issue.
- The proposed method, Antidote, shows significant improvement on hallucination mitigation, without introducing noticeable catastrophic forgetting.

**Weaknesses:**

### Major
- My biggest concern is that the fictitious presupposition questions (FPQs) are highly similar to MADBench [1].

MADBench reports that MLLMs “can be easily deceived by prompts with incorrect information”. For example, they ask the models “What color is the cat in the image?” with an image of dogs. The models will hallucinate a lot. This finding is highly related to the FPQs proposed in this paper.

Besides, they propose a benchmark MADBench, to evaluate MLLMs in deceptive questions, in 5 different categories. That is related to VFP-Bench in the paper.

I didn’t see citation to or comparison with MADBench in the paper. I feel it would be better if the authors:

(1) Discuss the relationship between their study and MADBench in the paper;

(2) Discuss the comparison between VFP-Bench and MADBench;

(3) Instead of claiming “They primarily … This paper highlights …”, the authors can just claim introducing a new benchmark and a new method to solve an already-found problem.

- When we ask FPQs to LVLMs, would some prompt strategies be able to help LVLMs to find the tricky part in the questions?

Maybe in the prompt, tell the model that the questions may contain something non-existent in the image?

Or use Chain-of-Thought prompting? For example, when we want to ask the model “what is the brand of the car”, first let the model think “is there a car in the image?”  If a recent model is good at conventional hallucination benchmark POPE as the authors claim, it should be improved by using this prompting strategy.

- I’m wondering if the preference optimization is necessary in Antidote. What if the authors just use supervised fine-tuning, taking the corrected responses as the ground truth?
- I’m concerned with the generalization ability of Antidote, which may be a common problem of existing hallucination mitigation methods. LVLMs are expected to do a wide range of tasks, like medicine, chemistry, and math. When image or task domains are changed, will the current methods be effective?

For example, LVLMs may be prompted to do Chart understanding tasks. In this case, there are chart-related FPQs, such as “what is the value of Australia” (there is no Australia in the chart). Antidote, relying on image generative model, may fail to generalize to different domains.

### Minor
- How do the authors collect questions on VFP-Bench? Are the questions generated by LLM or humans?
- Only LLaVA 1.5 (7B and 13B) and LLaVA 1.6 (7B and 13B) are covered in the experiments. These models are developed by the same team and are similar in design and training. So I’m concerned that the experiments cannot show the generalization of Antidote to different LVLMs.
- There is a very light “0” in the middle of “Large VLMs” in Figure 4.




----------------
[1] How Easy is It to Fool Your Multimodal LLMs? An Empirical Analysis on Deceptive Prompts. https://arxiv.org/abs/2402.13220.

**Questions:**

Please see the Weaknesses section.

---

### Official Review · Reviewer_qv8P · 2024-11-04

**Soundness:** 2
**Presentation:** 3
**Contribution:** 2
**Rating:** 5
**Confidence:** 4

**Summary:**

This paper presents a hallucination benchmark VFP-Bench and a relevant solution antidote. Even though the existing LVLM can identify the existence of objects in the images, they tend to fail in fictitious presupposition questions (FPQ). The paper constructs the VFP-Bench benchmark. Moreover, it proposes a synthetic data-driven self-correction post-training method, antinode, for corresponding hallucination mitigation through an automated data synthesis pipeline. Experimental results show that antinode not only reduces the FPQ-related hallucinations but also retains a certain level of general LVLM ability.

**Strengths:**

- The paper focuses on a specific type of hallucination, fictitious presupposition questions, which is intriguing. It defines the FPQ task well and proposes a benchmark.
- The benchmark consists of a balanced portion for FPQ and TPQ, eliminating favoring over-corrected or biased LVLM performance.
- It proposes a comprehensive pipeline for data synthesis as the self-correction solution for hallucination mitigation. A "factual assessor" scheme is adopted to ensure hallucination during synthesis.
- Multiple backbones are experimented across benchmarks. The enhancement on VFP-Bench is significant.

**Weaknesses:**

- The benchmark collection details are unclear and might not be well-designed. CC3M contains around 3M images, but VFP "carefully curated a set of 1,000 samples" (line 707) under an unrevealing policy. The question crafting process is also unclear. It is confusing that they notice four types (item, knowledge, scene, and activity) of questions for a benchmark at a scale of 1000, but the portion for each type is still uneven (from 8.9% to 39.0%). A relatively small benchmark is more acceptable when it involves massive efforts and is designed carefully.
- The paper states their method "does not rely on any expert models (e.g., GPT-4V)" (line 149), but it seems that the data generation pipeline relies on a large language model and an image generator. Both can introduce biases into this paradigm.
- It is doubtful whether the improvement is from training on the proposed synthetic data or a simple self-correction process. As shown in Figure 1, LVLMs are able to identify the absence of objects. The reason that LVLMs fall short in FPQ may be that they are misled by the FPQ questions. Intuitively, reformatting (i.e., prompting) FPQ into the POPE-type questions is a straightforward and potentially effective self-correction strategy. Given the complexity of the Antidote pipeline, the self-correction through prompting should be considered. Ideally, it may perform similarly to the POPE benchmark (Figure 1 b), which is higher than the Antidote-corrected one.
- Given the evaluation is conducted by the GPT-4o, it is necessary to validate the evaluation accuracy of GPT-4o.

**Questions:**

- What are the image curation criteria for VFP? How are they annotated (e.g., manually or automatically)? Is one type more important than the others, or why are the number of types so different in the benchmark?
- Without post-training on the synthetic data, what will the performance be with a simple prompting-based self-correction strategy? In other words, what will be the performance by prompting the LVLMs to interpret the FPQ as a POPE-type question?
- Could you provide experiments showcasing the accuracy of GPT-4o judgment?

---

### Note · Authors · 2024-11-13

I have read and agree with the venue's withdrawal policy on behalf of myself and my co-authors.